# INVARIANT RISKS WITHOUT KNOWLEDGE OF THE ENVIRONMENT

**Bratenkov Miron & Ivan Bondarenko**
Novosibirsk State University
Novosibirsk, Russia Federation
`{m.bratenkov, i.bondarenko}@g.nsu.ru`

## ABSTRACT

Generalization under data shifts is one of the challenges in machine learning. The paradigm of invariant risk minimization (IRM) provides an approach that can enhance the generalization capability of models when facing data shifts. Unfortunately, this approach is not without its drawbacks. One of the limiting factors for the widespread application of this paradigm is the requirement to partition the dataset into environments with different distributions. Dividing the dataset into the necessary subsets can often be problematic due to the complexity of the data. To address this issue, we propose a clustering-based approach that allows for the application of the IRM paradigm in any task, even without prior knowledge of the environments. Experiments demonstrate that using clusters as environments and training under the IRM paradigm on these environments improves the robustness of models to data shifts compared to training under the empirical risk minimization (ERM) paradigm. For the weather prediction task, the improvement was 10 percents in terms of the mean squared error (MSE) metric, while for the pre-training of a decoder model of a small language model (LLM), the increase was 75 percents on long texts according to the perplexity metric. Furthermore, the paper proposes a modification to the invariant risk minimization paradigms that simplifies the hyperparameter tuning for the penalty term of the error. This modification stabilizes the IRM training process and enhances the robustness of models compared to the traditional hyperparameter tuning for IRM. For the weather prediction task, the improvement was 10 percents in terms of MSE, and for the pre-training of the decoder model of an LLM, the increase was 460 percents on long texts in terms of perplexity compared to classical IRM.

## 1 INTRODUCTION

Over the past decade, machine learning technologies have rapidly evolved, facilitating advancements in computer vision (He et al. (2016)), speech recognition (Graves et al. (2013)), and numerous other fields (Bahdanau et al. (2015), Fu et al. (2019), Luong et al. (2015)). However, deeper investigations into neural network models and their learning mechanisms have revealed the shortcomings of these approaches due to the presence of spurious correlations in trained models (Beery et al. (2018a), DeGrave et al. (2021), Geirhos et al. (2020), Zhang et al. (2021)). Spurious correlations indicate the dependency of machine learning systems on the training data. They degrade the generalization ability on out-of-distribution data.

As a thought experiment, consider the task of classifying images of cows and camels (Arjovsky et al. (2019), Beery et al. (2018b)). To address this task, we label images of both animal species. The training data are collected such that the majority of cow photographs are taken in green meadows, while camel photographs are predominantly captured in deserts. In contrast, the test data are collected so that images of cows are taken on sandy beaches, while those of camels are taken in areas with abundant vegetation. Consequently, after training a neural network on this dataset, we will observe that the model fails to classify simple examples of cows and camels from the test data. Upon investigating the cause, we realize that the neural network has learned to classify cows and camels based on spurious correlations: cow - meadow, camel - desert. Thus, in this case, the background serves as a spurious, non-invariant feature that can change.

The primary paradigm of machine learning, Empirical Risk Minimization (ERM), is based on the assumption that all data are independent and identically distributed, which is not always the case. Models that minimize empirical risk can significantly degrade prediction quality if the test data differ substantially from the training data in distribution. This issue is known as out-of-distribution (OOD) generalization. To loosen the assumption of independence and identical distribution of data, the principle of invariance has been proposed [19]. This principle aims to utilize invariant representations of data, which remain stable even in the presence of distribution shifts. In the example with cows and camels, the invariant feature was the shape of the animals.

The Invariant Risk Minimization (IRM) paradigm (Arjovsky et al. (2019)) extends the principle of invariance to neural networks. The assumption of independence and identical distribution is replaced with the assumption that the data are divided into environments and that the spurious correlations of the data vary depending on the environments, while the invariant correlations remain stable. IRM aims to train neural networks to extract invariant data correlations while excluding spurious ones. It has considerable potential, as evidenced by recent studies in this area (Ahmed et al. (2020), Ahuja et al. (2020a), Ahuja et al. (2020b), Chang et al. (2020), Krueger et al. (2021), Rosenfeld et al. (2020)).

In this work, we consider a Bayesian modification of Invariant Risk Minimization (BIRM) (Lin et al. (2022)) as a method for minimizing invariant risks. Unlike IRM, which is inefficient for deep neural networks (Gulrajani & Lopez-Paz (2021), Lin et al. (2021)) due to its tendency to overfit, BIRM has been developed to address this shortcoming while maintaining a focus on discovering invariant representations during training. BIRM employs a Bayesian approach to IRM (Bernardo & Smith (2009)).

Unfortunately, the paradigms for minimizing invariant risks are not widely adopted, primarily because the data must be separated into environments. Properly dividing data into environments is not always feasible. Environments can be sufficiently complex and contain numerous parameters, making intuitive separation difficult, and there may not be a domain expert available to aid in this process.

The objective of this paper is to develop an approach that simplifies the application of IRM methods (including BIRM). The main tasks are as follows:

- Relax the requirement for data to be divided into environments, extending the paradigm of invariant risk minimization to tasks where the ERM paradigm is applicable.
- Simplify hyperparameter tuning for the IRM paradigm.
- Propose a model of environment for environment separation of data.
- Conduct a comparison of the proposed approach with ERM.

The modifications made to the invariant risk minimization algorithm have significant practical implications. In real-world tasks, models are trained on data that do not encompass the entire population of possible data for the task and are used on datasets not present in the training sample. Data shifts in such scenarios are present from the outset of model deployment, making robustness against such shifts especially important.

## 2 RISK MINIMIZATION PARADIGMS

### 2.1 EMPIRICAL RISK MINIMIZATION

We consider the general formulation of the supervised learning problem. Let there be two object spaces $X$ and $Y$ that are related by a joint distribution $P(x, y)$ over $X$ and $Y$. The objective is to find a function $h : X \to Y$ that associates object $x$ with object $y$. To identify the function $h$, we have a set of training data drawn from $P(x, y)$, represented as pairs $(x_1, y_1), ..., (x_n, y_n)$, where $x_i \in X$ are the input variables and $y_i \in Y$ are the corresponding outputs associated with $x_i$. It is also assumed that there exists a non-negative real-valued loss function $L(\hat{y}, y)$ that measures the discrepancy between the prediction $\hat{y}$ and the true value $y$. The risk associated with the function $h(x)$ is defined as the expected value of the loss function:

$$R(h) = E[L(h(x), y)] = \int L(h(x), y) dP(x, y).$$

The ultimate goal of the learning algorithm is to find the function $h^*(x)$ from a fixed class of functions $H$ such that the risk $R(h)$ is minimized:

$$h^* = \arg\min_{h \in H} R(h).$$

In general, the risk $R(h)$ cannot be computed because the joint distribution $P(x, y)$ is unknown. However, since the sample elements of the training data are independent and identically distributed, we can estimate the risk by computing the expected value with respect to the empirical measure:

$$R_{empirical}(h) = \frac{1}{n} \sum_{i=1}^{n} L(h(x_i), y_i).$$

The principle of empirical risk minimization (ERM) states that the learning algorithm should select the function $\hat{h}$ that minimizes the empirical risk for the class of functions $H$:

$$\hat{h} = \arg\min_{h \in H} R_{empirical}(h).$$

Thus, the learning algorithm defined by the empirical risk minimization paradigm involves solving a specified optimization problem.

The primary issue concerning out-of-distribution (OOD) generalization of the training set is that the function $\hat{h}$ obtained does not account for potential differences in distribution between the test data and the training data. Therefore, if the joint distribution $P^*(x^*, y^*)$ of the test data $X^*$ and $Y^*$ significantly differs from the joint distribution $P(x, y)$ of the training set $X$ and $Y$, the function $\hat{h}$ derived through ERM may perform considerably worse in predicting $y^*$ from input variables $x^*$ (example in Chapter 5.1, (Arjovsky et al. (2019))).

## 2.2 Invariant Risk Minimization

We now modify the problem formulation. Let the training data $X$ and $Y$ be partitioned into environments $e \in \mathcal{E}$ such that within all environments the data $X^e$ and $Y^e$ have different joint distributions $P(x^e, y^e)$. The objective of the invariant risk minimization learning paradigm is to discover correlations that are invariant across any learning environment. For prediction tasks, this means finding a representation of the data such that the optimal classifier on this representation is the same across all environments.

**Definition:** *A data representation function $\Phi : X \to H$ forms an invariant predictor $w \circ \Phi$ for environments in $\mathcal{E}$ if there exists a classifier $w : H \to Y$ that is optimal for all environments in $\mathcal{E}$, i.e., $w \in \underset{\hat{w}:H \to Y}{argmin} R^e(\hat{w} \circ \Phi)$ for any $e \in \mathcal{E}$.*

In this definition, the representation $H$ of the input data $X$ is an invariant representation, independent of the environments.

Thus, the optimization problem corresponding to the invariant risk minimization paradigm can be expressed as follows:

$$\min_{\substack{\Phi:X \to H \\ w:H \to Y}} \sum_{e \in \mathcal{E}_{train}} R^e(w \circ \Phi),$$

$$\text{where } w \in \arg\min_{\hat{w}:H \to Y} R^e(\hat{w} \circ \Phi) \quad \text{for all } e \in \mathcal{E}_{train}.$$

The resulting problem is a complex two-level optimization task, as each constraint invokes an internal optimization procedure. Consequently, the authors in (Arjovsky et al. (2019)) proposed a practically applicable version:

$$\min_{\Phi:X \to Y} \sum_{e \in \mathcal{E}_{train}} R^e(\Phi) + \lambda \cdot ||\nabla_{w|w=1.0} R^e(w \circ \Phi)||^2,$$

where $\Phi$ is the complete invariant predictor, $w = 1.0$ is a scalar and fixed "dummy" classifier, $||\nabla_{w|w=1.0} R^e(w \circ \Phi)||^2$ is a penalty term measuring the optimality of the dummy classifier in each environment $e$, and $\lambda \in [0, \infty)$ is a regularization hyperparameter that balances between predictive

performance (ERM) and the invariance of the predictor $1.0 \circ \Phi$. The expression can be compactly written as:

$$\min_{\Phi: X \to Y} R(\Phi, \mathcal{E}_{train}) + \lambda \cdot Penalty.$$

This notation is preferable for further discussion, as it is general for various formulations of IRM, including BIRM (Lin et al. (2022)).

As demonstrated in works (Lin et al. (2022), Vapnik, Chang et al. (2020)), the original formulation of IRM is not effective for deep models. In the case of deep models, when training under the IRM paradigm, there is a tendency toward overfitting. It has also been shown in (Lin et al. (2022)) that IRM can eventually degenerate into ERM, and that models trained using IRM may still contain spurious, non-invariant correlations in the data.

In addition, during experiments conducted as part of this research work, difficulties in using IRM were identified. This stems from challenges in selecting the hyperparameter $\lambda$. The problem will be discussed in more detail in Chapter 4.

Furthermore, it is worth noting the requirement for partitioning the training data into environments. Correct partitioning is not always feasible due to a lack of expertise or appropriate knowledge, and also due to the complexity and volume of the data. This renders the application of the IRM method impractical. This issue will be elaborated upon in Chapter 3.

## 2.3 BAYESE INVARIANT RISK MINIMIZATION

To address the issue of the inefficiency of IRM for large models, the authors in (Lin et al. (2022)) developed the BIRM method.

BIRM is a modification of the IRM method in which Bayesian theory has been applied. Let $h_u(\cdot)$ and $g_w(\cdot)$ denote the representation function for the data and the classifier corresponding to parameters $u$ and $w$, respectively. We define $D_u^e = \{h_u(x_i^e), y^e\}_{i=1}^n$ and $D_u = \bigcup_{e=1}^{\mathcal{E}_{train}} D_u^e$ as the data sets for environments after the transformation $h_u$ for the environment and the mixture of environments. The distributions $q_u(w) \approx p(w|D_u)$ and $q_u^e(w^e) \approx p(w^e|D_u^e)$ represent the posterior values of the classifiers accounting for the data transformation. The negative log-likelihood function for the data from environment $e$ is given by $R^e(w, u) = -\ln p(D^e|w, u) = -\sum_{i=1}^{n_e} \ln p(y_i^e|w, h_u(x_i^e))$.

Thus, the optimization problem for the paradigm of Bayesian Invariant Risk Minimization (BIRM) can be formulated as:

$$\max_u \sum_e \mathbb{E}_{q_u(w)}[\ln p(D^e|w, u)] + \lambda(\mathbb{E}_{q_u(w)}[\ln p(D^e|w, u)] - \mathbb{E}_{q_u^e(w^e)}[\ln p(D^e|w^e, u)]),$$

subject to

$$\mathbb{E}_{q_u(w)}[\ln p(D^e|w, u)] = \int \ln p(D^e|w, u) q_u(w) dw$$

$$\mathbb{E}_{q_u^e(w^e)}[\ln p(D^e|w^e, u)] = \int \ln p(D^e|w^e, u) q_u^e(w^e) dw^e,$$

where the expectations denote the logarithmic likelihoods $q_u(w)$ and $q_u^e(w^e)$ for the data from environment $e$, respectively.

The first term in the stated optimization problem maximizes the expected logarithmic probability of the overall posterior $q_u(w)$ by optimizing over $u$. The second term requires the examination of invariant correlations. If $h_u(\cdot)$ encodes non-invariant features, then the transformed distribution $D_u^e$ varies depending on the environment. A penalty is then imposed, which demands that $h_u(\cdot)$ discard non-invariant functions.

It is worth noting that the original definition of IRM is based on point estimates of parameters $w$, which can be highly unstable when data is scarce. Instead of point estimation, BIRM directly interacts with posterior distributions, making it less prone to overfitting (Lin et al. (2022)).

Due to this modification, BIRM is effective in cases of limited data or large models. However, two other inherent problems of IRM remain. First, as with IRM, it is still necessary to partition the data into environments for training under this paradigm. Second, the hyperparameter $\lambda$ continues to play a crucial role in the paradigm, as it regulates the extent of the penalty's influence during training, yet its selection remains a laborious task.

## 3 CLUSTER AS A MODEL OF ENVIRONMENT

### 3.1 ENVIRONMENT IN IRM

As previously mentioned, the application of Invariant Risk Minimization (IRM) paradigms has a significant requirement for the existence of environment partitioning. This requirement limits the application of the IRM paradigm, particularly in tasks with a large number of parameters or in conditions where expert knowledge in the domain is lacking, making it challenging to implement such partitioning.

In the context of the invariant risk minimization paradigm, an environment should be understood as a dataset in which the features of the environment or combinations of features have same distribution. Features of the environment or combinations of features are those that can introduce spurious correlations in the model's predictions.

Returning to the example of cows and camels (Arjovsky et al. (2019), Beery et al. (2018b)), the environment feature can be considered as the background, specifically a meadow or a desert. Thus, the dataset can be partitioned into one environment containing images of animals against the background of a meadow and another environment containing images against the background of a desert. This partitioning is intuitively clear; however, as will be demonstrated in the experimental section, more complex task examples will arise where intuitive partitioning becomes difficult.

### 3.2 CLUSTER AS A MODEL OF ENVIRONMENT

To address this issue, we propose partitioning the dataset based on clustering. As is well known, clustering is the task of grouping a set of objects into subsets (clusters) such that objects within the same cluster are similar concerning certain feature sets that distinguish depending on the clustering algorithm, and different from objects in other clusters based on these sets of features.

To satisfy the requirement of distinct environments based on distribution, in addition to clustering, statistical tests should also be utilized. According to the chosen statistical test, clusters should be merged when the main hypothesis holds. Thus, the clusters obtained after merging will have different distributions and can be utilized as environments in the paradigms of invariant risk minimization.

The partitioning of data into clusters and the use of clusters as environments allow for the application of the IRM learning paradigm when environments are not given or when independent partitioning becomes challenging. The results in subsections 5.2-5.3 demonstrate that partitioning data into environments based on clustering can improve the robustness of the model to data shifts and reduce the influence of spurious features on the model.

In this work, the k-means clustering algorithm was employed. It aims to minimize the total squared deviation of cluster points from their centers:

$$V = \sum_{i=1}^{k} \sum_{x \in S_i} (x - \mu_i)^2,$$

where $k$ is the number of clusters, $S_i$ are the derived clusters for $i = 1, 2, ..., k$, and $\mu_i$ is the centroid of all vectors $x$ in cluster $S_i$.

This clustering algorithm was chosen due to its efficiency for large datasets and the constraints of computational resources.

## 4 ADAPTIVE HYPERPARAMETER

### 4.1 HYPERPARAMETER SELECTION PROBLEM

As previously mentioned, the use of the Invariant Risk Minimization (IRM) paradigm presents challenges in selecting the weight coefficient of the penalty term in the loss function. This hyperparameter is crucial for effectively learning invariant features, as it influences the penalty term that discourages the model from training on non-invariant features.

## 4.2 Gradient Analysis of Errors

Let us consider the gradients (Fig. 1) that arise during the backpropagation of errors at the last layer of the neural network for the task discussed in subsection 5.1. As can be observed, the gradients of

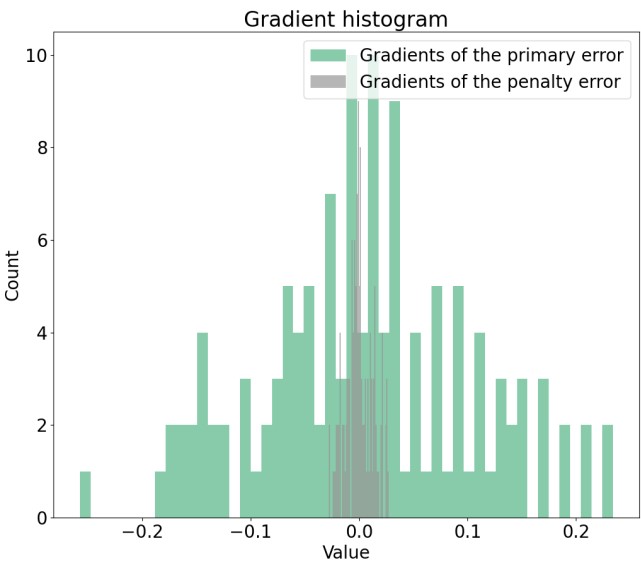

Figure 1: Histogram of gradients at one of the epochs

the primary loss exhibit a larger variance compared to the gradients of the penalty term, which are concentrated around zero. This distinction provokes an uneven impact on the training process. If the weight hyperparameter is static and equal to one, the model's weights will be more influenced by the gradients of the primary loss than by those of the penalty term.

This observation has prompted a consideration of the distributions of gradients over the entire training interval. As can be observed in Fig. 2, the distributions of the gradients are not static; they vary over time, with a significant difference in deviations from zero for the penalty term gradients and gradients of the main error persisting throughout.

Based on this data, two conclusions can be drawn. First, a static weight coefficient will be ineffective due to the variability of the last layer's gradient distributions. Second, the selection of a piecewise plan for adjusting the weight coefficient during training poses a challenging task, as this requires empirical tuning of the values based on a series of prior experiments.

## 4.3 Adaptive Hyperparameter Selection Algorithm

The adaptive weight coefficient selection algorithm replaces the weight hyperparameter with a decay coefficient, the tuning of which is less labor-intensive.

$$L(\theta) = L_u(\theta) + \lambda \cdot Penalty(\theta),$$
$$D(\nabla_\theta Penalty(\theta_n)|_{\text{last layer}}) = \sigma_p,$$
$$D(\nabla_\theta L_u(\theta_n)|_{\text{last layer}}) = \sigma_u,$$
$$\lambda = \frac{\sigma_u}{\sigma_p},$$
$$\lambda_n = \alpha \cdot \lambda_n + \lambda.$$

To employ this algorithm, it is required that the weights be initialized from a distribution with zero mean and that the activation functions equal zero at the origin. When initializing weights with a

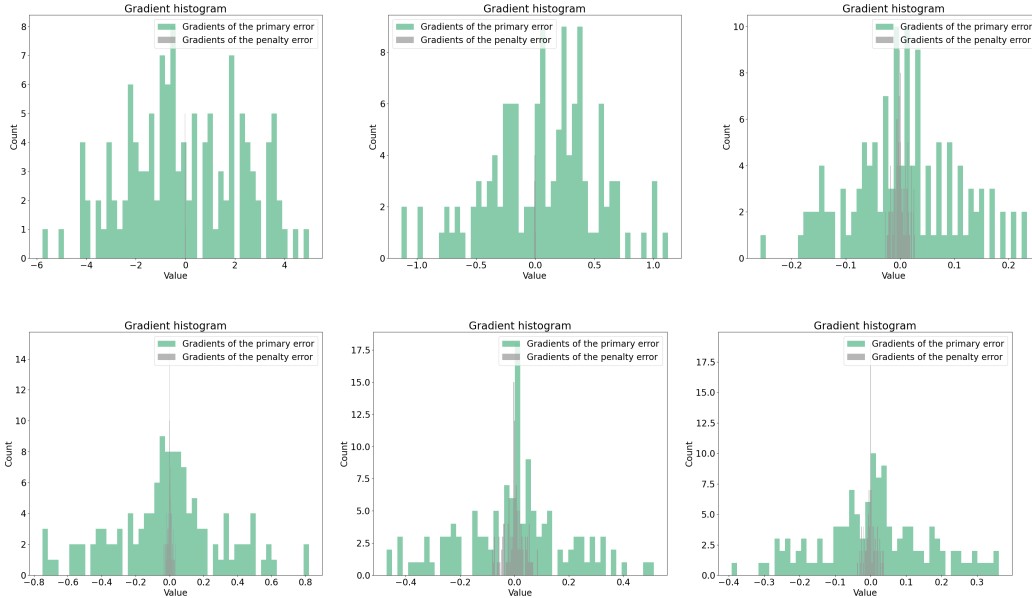

Figure 2: Histograms of gradients at epochs 0, 200, 400, 600, 800, and 1000, respectively.

zero mean, the gradients during backpropagation will also have a zero mean. Thus, multiplying the penalty error by the weight coefficient will not affect the mean of the gradients, only their variance.

Let us denote the weights of the neural network at the $n$-th layer as $w_{ij}^n$, the variance as $\mathcal{D}[w_{ij}^n] = \sigma_n$, and the input vector to this layer as $z^n$. The activation function is denoted as $\phi(x)$, with the argument being the row vector $s^n$. Then, we can express the expectation as follows:

$$\mathbb{E}[\frac{\delta L}{\delta s_j^n}] = \sum_{k=1}^{d_n+2} \mathbb{E}[L_k^{n+1} w_{jk}^{n+1}] = \sum_{k=1}^{d_n+2} \mathbb{E}[L_k^{n+1}] \underbrace{\mathbb{E}[w_{jk}^{n+1}]}_{=0} = 0.$$

The most popular weight initialization algorithms are Xavier Glorot and He Kaiming methods. For these algorithms, weights have a zero mean. They are also standard in libraries such as Torch and TensorFlow. Additionally, for self-normalizing neural networks, zero-mean weight initialization is employed, indicating that the proposed algorithm is applicable to this type of neural network architecture as well.

## 4.4 Final Algorithm

The final algorithm is structured as follows:

1. **Data Preprocessing**. In this phase, the available data is further processed. For example, it may be filtered to remove outliers, normalized, or subjected to dimensionality reduction.

2. **Data Clustering**. The data is clustered using a suitable clustering algorithm.

3. **Cluster Filtering**. Clusters with similar distributions are merged. Statistical tests are employed to assess the distributions of the clusters.

4. **Cluster as a Medium Model**. The data is divided into training, validation and test sets. Additionally, for the training data, clusters are equated to enviroments, facilitating the subsequent application of IRM methods.

5. **Training**. Training is conducted according to the paradigms of IRM, utilizing an adaptive hyperparameter tuning algorithm.

## 5 EXPERIMENTS

### 5.1 SYNTHETIC DATASET

The synthetic dataset is defined as follows:

$$\begin{cases} X_{inv} \sim N(E_2, I_2) \\ Y = 1^T \cdot X_{inv} + N(0, 0.1) \\ X_{env} = Y + N(E_2, pe \cdot I_2) \\ X = (X_{inv}, X_{env}), \end{cases}$$

where $X_{inv}$ represents the invariant variables, $Y$ is the target variable, $X_{env}$ denotes the environmental variables, $E_2$ is the zero matrix of dimension 2, $I_2$ is the identity matrix of dimension 2, $pe$ is the environmental parameter, and $X$ constitutes the input data.

The training dataset included environments with the following parameters:

$$pe_{train} = [0.1,\ 0.3,\ 0.5,\ 0.7,\ 0.9]$$

The validation dataset encompassed environments with parameters:

$$pe_{val} = [0.4,\ 0.8]$$

The test dataset involved environments with parameters:

$$pe_{test} = [10,\ 100]$$

The quality metric employed was the ratio of the mean squared error (MSE) for data with environmental parameters $pe_{test}$ from the test distribution to that for data with parameters $pe_{val}$ from the validation distribution. Data from the test environment are referred to as $outdomain$, as they lie outside the distribution of the training dataset due to significantly different environmental parameters from those in $pe_{train}$. Conversely, data from the validation environment are termed $indomain$, as they possess a similar distribution to the training dataset, with environmental parameters that closely align with $pe_{train}$.

$$Metrica = \frac{MSE_{out\,domain}}{MSE_{in\,domain}},$$

This metric will indicate the factor by which the mean squared error on the training distribution data differs from that on the out-of-distribution data.

This example does not utilize data splitting into environments, as it aims to demonstrate the importance of hyperparameter tuning. The main idea of the synthetic dataset is as follows. Since the input variables follow a normal distribution with a zero mean, the minimum mean squared error will be achieved if the model predicts the mean, which is zero. Given that the loss function employed is the mean squared error, the minimum possible value of the loss function depends on the variances of the invariant variables $X_{inv}$, the environment parameter $pe$, and the noise level of $0.1$. If the variance of the sum of the invariant variables and noise exceeds $pe$, the minimum possible value of the loss function will equal $pe$. Otherwise, the minimum possible value of the loss function will equal the sum of the variance of the invariant variables and the noise.

In the training dataset, $pe$ is always less than the variance of the sum of the invariant variables and noise, allowing the model to learn to predict the target variable based on the environment variables, i.e., relying on spurious correlations rather than invariant ones. Consequently, on the test dataset, due to the high values of $pe$, the model trained on environment variables will significantly degrade in performance.

Let us consider the obtained results. As shown in the histogram (Fig. 3), the model trained under the ERM paradigm is highly susceptible to the effects of data shift. In contrast, the BIRM model with a static hyperparameter equal to 1 exhibits greater robustness to data shifts. When applying adaptive hyperparameter tuning, the differences between predictions on data with a distribution similar to the training dataset and out-of-distribution data are less than twice as large, indicating a performance improvement of 240 times based on the selected metric compared to ERM, and 73 times better than BIRM with a static hyperparameter. According to Table 1, one might perceive

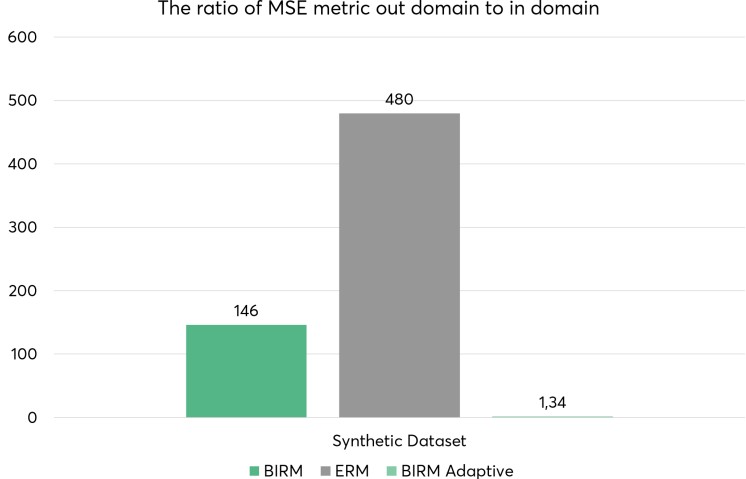

Figure 3: Histogram of the ratio of the mean square error of data outside the distribution to data with the distribution of the training sample for models trained using the ERM, BIRM, and BIRM paradigms with an adaptive hyperparameter.

Table 1: The values of the root mean square error for different methods in domain and out domain samples.

| Data | ERM | BIRM | Adaptive BIRM |
|---|---|---|---|
| In domain | 0.01 | 0.022 | 1.7 |
| Out domain | 4.8 | 3.2 | 2.3 |

that methods for minimizing invariant risks significantly deteriorate the model's predictive capacity, and that the comparison of robustness to data shifts is invalid. However, this is not the case. As will be demonstrated below, there will be no such substantial differences for models trained under the ERM and IRM paradigms. Furthermore, models trained under IRM will maintain superiority in robustness to data shifts for more applied tasks. This result merely illustrates that, in the absence of complete information about all possible data variations, models trained based on ERM principles may yield unexpectedly poor predictive quality on unseen data, whereas the predictive quality of models trained according to IRM principles will be more anticipated.

## 5.2 Weather Prediction

In this study, the application of the proposed algorithm was investigated in scenarios where the partitioning into environments is unknown a priori. The task at hand involved predicting the temperature at a height of 2 meters based on 123 meteorological parameters. The dataset was sourced from the Shifts Challenge 2019 competition, which primarily focused on the data shift problem. The complete dataset consisted of 15 million examples, with 3 million allocated for testing and validation. Within the competition, the test data was divided into two groups, namely in-distribution and out-of-distribution samples. The out-of-distribution data included samples from snowy and polar climates collected after May 14, 2019. The training data consisted of tropical, humid, and temperate climate data collected from September 1 to April 8, 2019. Thus, the data shift occurs due to both the temporal aspect of data collection and the climatic conditions.

In the proposed algorithm, the data were clustered into 30 clusters using the K-means clustering algorithm. Subsequently, the distributions of the obtained clusters were compared using the multivariate Mann-Whitney criterion and merged if the null hypothesis was not rejected. Ultimately, 27 clusters with distinct distributions were obtained, which were subsequently utilized as environments for the paradigm of minimizing invariant risks.

Table 2: Values of the root mean square error for different methods on the train, in domain and out domain

| Data | ERM | BIRM | Adaptive BIRM |
|---|---|---|---|
| Train | 0.36 | 0.38 | 0.43 |
| In Domain | 0.36 | 0.39 | 0.43 |
| Out Domain | 0.48 | 0.49 | 0.49 |

The quality metric employed was derived from the previous task.

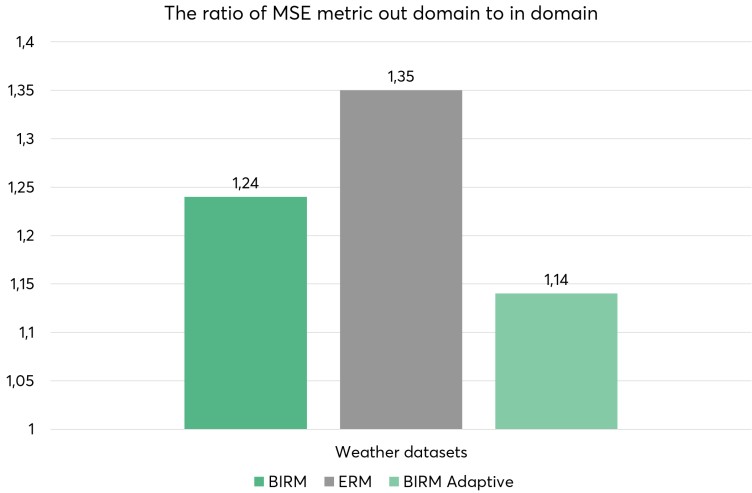

Figure 4: Histogram of the ratio of the mean squared error of the data out of distribution to the data with the training sample distribution for models trained using the ERM, BIRM, and BIRM paradigms with an adaptive hyperparameter.

As illustrated in the histogram (Fig. 4), the model trained under the ERM paradigm is more susceptible to data shifts than the other models. Meanwhile, the model with BIRM using a static hyperparameter equal to 1 shows improvement over the ERM model, consistent with previous findings. When applying adaptive hyperparameter tuning, the resulting model exhibits even greater resilience to data shifts.

According to Table 2, consistent with the synthetic dataset, an increase in the root mean square error is observed for the BIRM methods, along with a decrease in the discrepancy between the errors for in-distribution and out-of-distribution datasets. In contrast to the synthetic task, the differences in root mean square error among the various methods are not as pronounced. It is also worth noting that the increase in root mean square error in this case does not indicate a failure of the IRM methods. This increase may be attributed to the existing data not covering the full scope of potential values and combinations, and the invariant correlations identified by the IRM models yield only such error values. The improvement of the expected prediction is crucial for resilience to data shifts, and the Adaptive BIRM model demonstrates the best performance, as the ratio of error values is minimized.

## 5.3  TEXT GENERATION TASK

In light of the popularity of large language models such as Chat GPT, Gemini, and others, the task of text generation has been explored. The Taiga dataset was utilized as the textual dataset. This dataset comprises a corpus consisting of 77% fictional texts (33 literary magazines), 19% poetry, 2% news articles (from 4 popular websites), and 2% other texts (popular science, cultural magazines, social media, amateur websites).

In text generation tasks, one possible parameter defining the data shift is the length of the generated text. This parameter was chosen as the data shift in this study. The data were clustered using the K-means clustering algorithm into three clusters based on the length of token sequences. As a result of clustering, the median number of tokens in the three clusters amounted to 46, 182, and 898 tokens, respectively. Subsequently, the clusters with the minimum and average lengths were used for training as the means, while the cluster with the largest average length was utilized as out-of-distribution data.

The following datasets from the Taiga corpus were employed in the task: arzamas, interfax, kp, lenta, nplus1, proza ru, fontanka, social, stihi ru. Data from each dataset were collected in approximately equal amounts, amounting to 200,000 examples. A text example was extracted from the dataset and, if it contained multiple sentences, it was split accordingly. Both the original text example and the sentence breakdown were included in the training dataset if applicable. Subsequently, all examples were tokenized and truncated to a maximum of 1000 tokens.

Afterwards, the process of preparing the means and splitting the dataset into training, testing, and validation sets took place. The K-Means clustering method based on the length of tokenized examples was used for mean extraction. According to the aforementioned approach, clusters with average and minimum average lengths of examples were included in the training set. The validation set comprised 2000 examples from each cluster of the training set (which were not included in the training process), while the test set included 2000 examples from all three clusters (with these examples also excluded from training). Ultimately, the training set comprised 1,600,000 examples, the validation set contained 4000 examples, and the test set included 6000 examples.

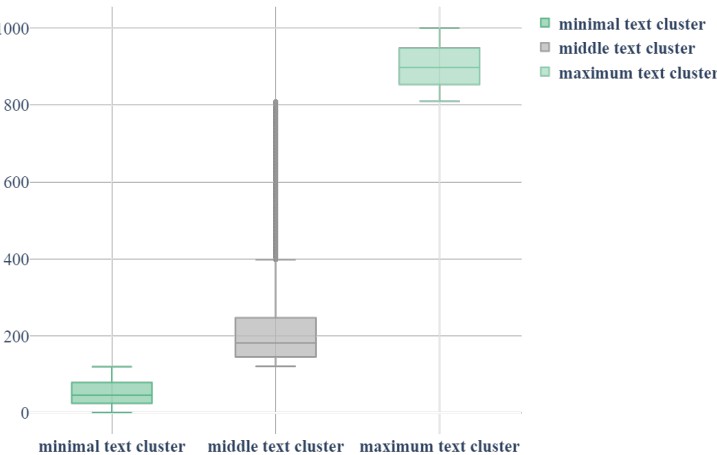

Figure 5: Histogram of data distribution by clusters with minimum, average and maximum length of text examples. The number of examples in a cluster with a minimum length of 1,100,000, with an average length of 500,000 and with a maximum length of 6,000

The quality metric employed was perplexity.

$$PPL(X) = \exp\left\{ -\frac{1}{t} \sum_{i=1}^{t} \log p(x_i | x_{<i}) \right\},$$

where $\log p(x_i | x_{<i})$ denotes the logarithmic likelihood of the $i$-th token given the tokens indexed by $< i$.

This quality metric reflects how confident the model is in its predictions; a lower perplexity value indicates that the model is attempting to generate more familiar text, while a higher perplexity value implies that the model has doubts about its predictions and that the predicted data are less familiar to it.

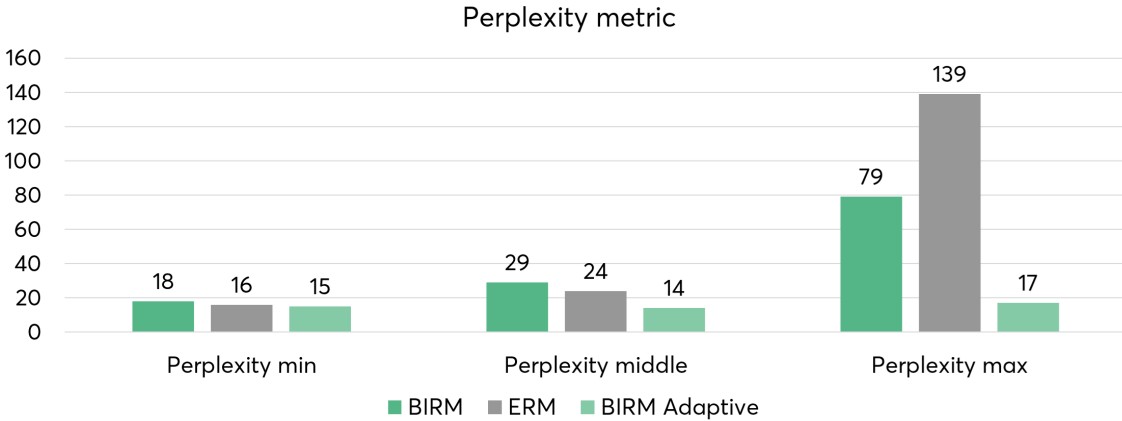

Figure 6: Histogram of perplexity values by clusters for models trained using the ERM, BIRM, and BIRM paradigms with an adaptive hyperparameter.

As evident in the histogram(Fig. 6), the model trained under the BIRM paradigm with an adaptive hyperparameter becomes robust to this type of data shift, as the perplexity values across the three clusters are virtually indistinguishable. Furthermore, this model exhibits the lowest perplexity among all investigated models. This result can be attributed to the fact that BIRM, as a paradigm, is aimed at training large, deep models on substantial datasets, as in this task.

# 6 FUTURE WORK

As demonstrated, applying invariant risk minimization paradigms to problems where environment partitioning is performed using cluster analysis and statistical tests enhances the robustness of models to data shifts. In this study, only one clustering algorithm, K-means, was considered due to real-world computational resource constraints. In future work, we plan to explore various data clustering methods in the context of environment discovery to understand their impact on robustness and how to select hyperparameters to achieve maximum stability with a specific clustering method.

Furthermore, we plan to develop an open-source library for applying various IRM paradigms in tasks involving tabular and text data. The library will include examples of its usage on the tasks examined in this paper.

# 7 CONCLUSION

This work presents two modifications of the invariant risk minimization (IRM) paradigms. The first modification extends the application of the paradigms to problems where partitioning into environments is not feasible, utilizing cluster analysis and statistical tests. In this modification, the environments are represented by a set of clusters, whose data exhibit different distributions. The second modification simplifies the application of the IRM paradigms. It is based on adaptive selection of the weight hyperparameter in the penalty error. The proposed algorithm for the adaptive weight hyperparameter in IRM enhances the robustness of models against data shifts.

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
