# OpenReview forum: "Invariant risks without knowledge of the environment"
_mathai.club/MathAI/2025/Conference — MathAI 2025 Oral_

### Official Review · Reviewer_ZYS4 · 2025-02-27
**The study presents an approach to applying Invariant Risk Minimization (IRM) without prior knowledge by leveraging data clustering, enhancing robustness to distribution shifts. However, the article needs clearer justification of the clustering algorithm choice and improved consistency in terminology and formatting.**

**Rating:** 6
**Confidence:** 3

**Review:**

Overview:
The study aims to remove the limitations on the applicability of the Invariant Risk Minimization (IRM) paradigm due to the need to partition data into environments with different distributions. In tasks where the data is complex or lacks expert knowledge for description, the requirement to partition the data complicates the use of IRM. An approach based on data clustering is proposed, which allows the application of IRM without prior knowledge of the environments, as well as a modified algorithm to simplify the tuning of hyperparameters. The technical approach is developed based on existing IRM and Bayesian IRM (BIRM) methods. Experiments confirm the effectiveness of the proposed approach on both synthetic and real-world tasks, demonstrating improved model robustness to changes in the data.
Relevance of the Topic:
The problem of generalizing machine learning models to data with distributions different from the training data (out-of-distribution generalization) is relevant in modern machine learning. Improving model robustness to data shifts is of great importance for practical applications such as weather forecasting, natural language processing, and other areas where data may change over time or depending on the context. The use of data clustering to describe environments is proposed, which allows the application of IRM even in the absence of explicit partitioning of data into environments. This expands the applicability of IRM to tasks where partitioning data into separate environments is difficult. An adaptive hyperparameter tuning algorithm for the penalty term in the IRM loss function is used, which simplifies the training process and enhances model robustness to data shifts. The proposed clustering-based approach can significantly expand the use of invariant learning in practice.
Recommendations for Article Revision:
•	It is advisable to justify the choice of the clustering algorithm and its parameters, as well as to discuss potential issues that may arise when selecting other clustering algorithms.
•	Consistency in terminology should be ensured: make sure that terms such as "data" are used in a uniform form (either "data are" or "data is"). Verb tenses should also be made consistent, especially in the description of experiments and methods.
•	In the sentence: "The training dataset included environments with the following parameters: [0.1;0.3;0.5;0.7;0.9]." — a semicolon is used to separate numbers, which does not conform to the standard format. Typically, a comma is used in such cases: [0.1,0.3,0.5,0.7,0.9].

---

### Official Review · Reviewer_4m3z · 2025-02-27
**The "Invariant risks without knowledge of the environment" paper can be accepted to the MathAI 2025 conference**

**Rating:** 8
**Confidence:** 3

**Review:**

This paper is devoted to solution of such important task as data shifts in machine learning. Use of improvements of Bayesian Invariant Risk Minimization method allows authors to solve this task.

This paper has the following disadvantages:
1) Authors should describe (in the paper) that invariant risk minimization paradigm handles the automatic detection of invariant features.
2) Authors should add short description of paper structure in the end of "Introduction" section to simplify understanding paper content.

---

### Official Review · Reviewer_DLnr · 2025-02-28
**Decent modification of BIRM algorithm, clear and solid experimental results.**

**Rating:** 7
**Confidence:** 4

**Review:**

Authors give a broad introduction to problem of enviroment identification and data shifts and describe the development of approaches to handle it step-by-step.

Advantages:

1.The descriprion of method is clear and transparent.

2.The experiments both on synthetic and real-world data demonstrate advantages of proposed novel method.

Disadvantages:

1.Low data&code availability(it would be interesting to see live implementations)

2.The role of clustering algorithms is unclarified(only K-Means used).

---

### Decision · Program_Chairs · 2025-03-08

**Decision:**

Accept (Oral)

**Comment:**

Your article has been accepted and you can make a presentation on the article. All articles will be sorted by rating and within the available conference places one author from each article will be invited. If there are not enough places, then you will either have the opportunity to present remotely or come at your own expense!